# Long-term exposure to particulate air pollution and black carbon in relation to natural and cause-specific mortality: a multicohort study in Sweden

Johan Nilsson Sommar [1], Eva M Andersson,[2] Niklas Andersson,[3] Gerd Sallsten,[2] Leonard Stockfelt,[2] Petter LS Ljungman,[3,4] David Segersson,[5] Kristina Eneroth,[6] Lars Gidhagen,[5] Peter Molnar,[2] Patrik Wennberg,[7] Annika Rosengren,[8] Debora Rizzuto,[9,10] Karin Leander,[3] Anton Lager,[11,12] Patrik KE Magnusson,[13] Christer Johansson,[6,14] Lars Barregard,[2] Tom Bellander,[3,15] Göran Pershagen,[3,15] Bertil Forsberg[1]

## ABSTRACT

**Objectives** To estimate concentration–response relationships for particulate matter (PM) and black carbon (BC) in relation to mortality in cohorts from three Swedish cities with comparatively low pollutant levels.

**Setting** Cohorts from Gothenburg, Stockholm and Umeå, Sweden.

**Design** High-resolution dispersion models were used to estimate annual mean concentrations of PM with aerodynamic diameter ≤10 µm (PM10) and ≤2.5 µm (PM2.5), and BC, at individual addresses during each year of follow-up, 1990–2011. Moving averages were calculated for the time windows 1–5 years (lag1–5) and 6–10 years (lag6–10) preceding the outcome. Cause-specific mortality data were obtained from the national cause of death registry. Cohort-specific HRs were estimated using Cox regression models and then meta-analysed including a random effect of cohort.

**Participants** During the study period, 7 340 cases of natural mortality, 2 755 cases of cardiovascular disease (CVD) mortality and 817 cases of respiratory and lung cancer mortality were observed among in total 68 679 individuals and 689 813 person-years of follow-up.

**Results** Both PM10 (range: 6.3–41.9 µg/m³) and BC (range: 0.2–6.8 µg/m³) were associated with natural mortality showing 17% (95% CI 6% to 31%) and 9% (95% CI 0% to 18%) increased risks per 10 µg/m³ and 1 µg/m³ of lag1-5 exposure, respectively. For PM2.5 (range: 4.0–22.4 µg/m³), the estimated increase was 13% per 5 µg/m³, but less precise (95% CI −9% to 40%). Estimates for CVD mortality appeared higher for both PM10 and PM2.5. No association was observed with respiratory mortality.

**Conclusion** The results support an effect of long-term air pollution on natural mortality and mortality in CVD with high relative risks also at low exposure levels. These findings are relevant for future decisions concerning air quality policies.

## INTRODUCTION

Particulate matter (PM) ambient air pollution exposure has been associated with increased risk of premature mortality in several cohort studies.[1 2] According to the Global Burden of Diseases Study 2017,[3] assessing 84 risk factors for premature mortality in 195 countries, ambient PM air pollution was ranked as the tenth most important risk factor for mortality and the most important environmental risk factor. Using risk functions restricted to cohort studies of outdoor air pollution, ambient particle matter pollution was estimated to cause 2.9 million (95% CI 2.5 to 3.4) deaths in year 2017.

In a review of 11 cohort studies on long-term air pollution exposure and mortality, ambient concentrations of PM with aerodynamic diameter ≤2.5 µm (PM2.5) were associated with increased risk of all-cause mortality and particularly cardiovascular mortality. However, the heterogeneity between study

**STRENGTHS AND LIMITATIONS OF THIS STUDY**

⇒ Few previous studies have followed up individuals by home address and time-varying exposures as annual mean concentrations at actual place of residence.

⇒ High-resolution dispersion models of particle concentrations were used to capture local contrasts in exposure.

⇒ The study did, however, not include information on noise exposure and green space near the home address.

⇒ This study showed associations with mortality within populations with comparatively low particle concentrations, generally below European Union standards.

⇒ As the first (multi)cohort study using high-resolution dispersion modelled particle concentrations at mean levels below WHO guidelines, this study found high relative risks of mortality.

BMJ

relative risk estimates was large.[2] The large European Study of Cohorts for Air Pollution Effects (ESCAPE) project including 22 cohorts, observed an increased risk of overall mortality associated with ambient PM2.5 concentrations[4] whereas no association was found with cardiovascular mortality.[5] In a recent study within a large Danish cohort found increased risks for both natural cause and cardiovascular disease (CVD) mortality associated with PM2.5 exposure during the last 15 years.[6] Neither the Danish study nor the ESCAPE study was able to demonstrate associations with respiratory mortality.[6 7]

Black carbon (BC), one component of PM, has been hypothesised to play a key role in the adverse health effects of air pollution exposure.[8] The review by Luben *et al* identified three studies on the long-term effect of BC (or elemental carbon) and mortality in coronary heart disease[9] and ischaemic heart disease.[10 11] However, the review did not find support for a stronger association with mortality for BC compared with PM2.5. Elemental carbon has been associated with all-cause mortality in another meta-analysis.[2] Within the ESCAPE project, PM2.5-absorbance was used as a proxy for BC. The estimated increased risk of natural mortality associated with PM2.5-absorbance did not reach statistical significance.[4] In Denmark (areas of Copenhagen and Aarhus), BC was, however, associated with both all-cause and cardiovascular mortality.[6]

These previous studies varied in the method used for the exposure assessment of particle concentrations. Earlier studies assessed concentrations using centrally located monitoring stations, mostly representing the urban background.[12] Later studies use air pollution monitoring together with data on land use (LUR) or detailed data on emissions and meteorological conditions (dispersion models). Most studies on long-term air pollution effects have assessed outdoor residential exposures at the year of inclusion into the cohort, a recent study on mortality however used annual mean concentrations on addresses during follow-up.[6] Studies using exposure assessments with a higher precision and in low-level exposure environments demonstrated higher relative risks of mortality in a recent meta-analysis.[13] Studies on low level air pollution are few, however. Adding to the knowledge from the long-term air pollution effects on mortality from the Canadian Community Health Survey-Mortality cohort[14–16] and a Medicare population in New England,[17] this study is of importance for policy making and determination of limit values. Compared with these previous studies that used a $1 \times 1 \, km^2$ resolution for the assessment of air pollutant concentrations this study will make use of high-resolution $50 \times 50 \, m^2$ dispersion modelled PM able to capture also local exposure contrasts.

## AIM

This study aims to use high-resolution dispersion modelled annual mean residential PM10, PM2.5 and BC concentrations to estimate associations with natural and cause-specific mortality in a northern European environment characterised by annual averages largely compliant with European Union (EU) air quality standards.

## METHODS AND MATERIALS
### Study cohorts

The study included four cohorts; two cohorts from Gothenburg, one from Stockholm (pooled from four cohorts) and one cohort from Umeå. Our study period ranged from 1 January 1990 to 31 December 2011 (31 December 2013 for Umeå participants residing at the same address as in 2011). Residential address history was obtained for all cohort participants through linkage using personal identification numbers to mandatory records of residential addresses at Statistics Sweden or the Swedish Taxation Authority. These residential addresses were then geocoded by matching against the Swedish Mapping Cadastral and Land Registration Authority databases. When needed, addresses were manually checked and corrected for inconsistencies and assigned geographical coordinates.

From Gothenburg, two general population cohorts were included: The Primary Prevention Study (PPS) cohort and the Multinational Monitoring of Trends and Determinants in Cardiovascular Diseases (GOT-MONICA) cohort. The PPS cohort recruited a random third of all men in Gothenburg born between the years 1915 and 1925 who were examined in 1970–1973 to study predictors of CVD.[18] The GOT-MONICA cohort is one of the cohorts within the international MONICA project designed to study risk factors for CVDs.[19 20] GOT-MONICA included residents aged 25–64 years. Recruitments were conducted in 1985, 1990 and 1995. For both cohorts, background data and information on cardiovascular risk factors were recorded by questionnaires. The recruited individuals were also examined by healthcare professionals.

Recruiting individuals from Stockholm County the Cardiovascular Effects of Air pollution and Noise Study (CEANS) cohort was included.[21] The cohort consists of four subcohorts. The Stockholm Diabetes Prevention Programme (SDPP) was designed as a population-based prospective cohort, and includes 3 128 men recruited between the years 1992 and 1994 and 4821 women recruited between the years 1996 and 1998. The programme recruited individuals aged 35–56 years that had not previously been diagnosed with diabetes. About half of the recruited had a first or second degree relative with a history of diabetes. These were then matched with recruits of the same age and sex, who did not have a relative with a diabetes history. The second subcohort named SIXTY consists of a population-based random sample of 60-year-old men and women (N=4 232) living within Stockholm County between August 1997 and March 1999. The third cohort consists of the Screening Across the Lifespan Twin study, including twins born before year 1958. Only participants residing in Stockholm County at recruitment were included in the current analysis (N=7

043). The fourth subcohort, the Swedish National Study on Aging and Care in Kungsholmen, included a random sample of inhabitants aged at least 60 years and residing in a central area of Stockholm City (N=3 363). Subcohort-specific characteristics of included individuals in CEANS have previously been presented.[22]

Including Umeå municipality, the Västerbotten Intervention Programme (VIP) is a programme where the population in the county is invited to a health examination the year they turn 40, 50 and 60 (and during some years also 30) years old.[23] The programme was initiated in parts of the county in 1985 as a community-level and individual-level programme to reduce the morbidity and mortality from CVD in Northern Sweden. The interview questions provide information about cardiovascular risk factors as well as social situation, education, diet and physical activity. So far more than 100 000 individuals have participated in the programme.

## Patient and public involvement

No patient involved. The public was also not involved in the design, or conduct, or reporting, or dissemination plans of the research.

## Mortality outcomes

Cause-specific mortality was determined by linkage of national personal identification numbers to the Cause of Death Register at the Swedish National Board of Health and Welfare. We used the International Code of Diseases (ICD)-9 001–779 or ICD-10 A00-R99 to define deaths by natural causes, ICD-9 400–440 or ICD-10 I10–I70 to define deaths in CVD, ICD9 162 or ICD10 C34 to define lung cancer deaths, and ICD9 460–519 or ICD10 J0–99 to define non-malignant respiratory mortality. In addition, deaths by other causes was studied (that is other than CVD, lung cancer, respiratory disease and external causes).

## Exposure assessment

An in-depth description of the dispersion model used for the exposure assessment has previously been published.[24] Both regional and local emission inventories for the years 1990, 2000 and 2011 in Gothenburg and Umeå, and years 1990, 1995, 2000, 2005 and 2011 in Stockholm, were used as input to Gaussian dispersion model simulations of annual mean concentrations of PM, that is, PM with aerodynamic diameter ≤10 µm (PM10), PM2.5 and BC. To allow high resolution in vicinity of roads, a quadtree receptor grid was used, resulting in a resolution down to 35 m x 35 m. For inner-city streets with buildings on one or both sides, an additional concentration component was simulated with the Danish operational street pollution model.[25]

Monitoring data were used in Stockholm to adjust the modelled concentrations to meteorological year-to-year variability. In Gothenburg and Umeå, interpolation was used between the years and each year was also adjusted by a ventilation factor taking meteorological variability

into account. Emission factors for traffic PM-exhaust for different vehicle types, speeds and driving conditions were calculated based on the Handbook on Emission Factors for Road Traffic V.3.1,[26] whereas emission factors for BC were based on TRANSPHORM.[27] The non-exhaust emissions of PM consist mainly of road wear particles, with a minor contribution from brake and tire wear.[28–30] For small-scale residential heating, emissions in Gothenburg and Stockholm were determined using household energy consumption data from Statistics Sweden with a resolution of 100×100 m. This was based on proxy data including number of stoves or boilers in each municipality, living space of small houses per square km, population density per 100 $m^2$ and availability of district heating. In Umeå a detailed inventory of individual stoves and boilers was used with data from chimney sweepers and interviews about amount of wood burning.[24] Industrial and energy production facilities were included as point sources in the model. Emission from shipping was also included using a method described by Jalkanen *et al*.[31]

To model the long-range transport of PM10, PM2.5 and BC the difference between total concentrations measured at monitoring stations and modelled local particle concentrations at the same location was considered on an annual basis (Gothenburg and Umeå). In Stockholm annual concentrations from a regional background station was used to estimate the long-range transport. The long-range transport contributions were estimated differently for each city, but within each city they had a purely temporal contrast. This was assumed since the spatial variation between regional background sites has been shown to be small.[32] These yearly contributions of background concentrations were then added to the estimated local concentrations. In a previous study, validations of total concentrations against measurements showed $R^2$ values of 0.87, 0.65 and 0.93 for PM10, PM2.5 and BC, respectively.[24]

The resulting concentrations were lastly added to each study participant's home addresses and annual exposures estimated accounting for any changes in address during the study period 1990–2011.

## Confounders

Cohort-specific models were adjusted for potential confounding by including sex, calendar year, subcohort (in Stockholm), smoking status (current, former, never smoker), alcohol consumption (in Stockholm and Umeå; daily, weekly, seldom, never), physical activity (in Gothenburg and Umeå: sedentary, moderate, intermediate or vigorous; in Stockholm: once a month or less/>1 hour per week, about once a month/1 hour per week, 3 times a week or more/>2 hours per week), marital status (single, married or living with partner, no answer), socioeconomic index by occupation (in Gothenburg (PPS) and Stockholm: blue collar, low and intermediate white collar and self-employed, high level white collar and self-employed professionals

with academic degrees, no answer), education level (in Gothenburg (MONICA), Stockholm and Umeå: primary school or less, up to secondary school or equivalent, university degree or more, no answer), occupation status (in Stockholm and Umeå: gainfully employed, unemployed/not gainfully employed, retired, no answer). Area-level socioeconomic status was estimated for each cohort member's residence from the mean neighbourhood individual income in persons of working age by Small Areas for Market Statistics provided by Statistics Sweden for the calendar year 1994.

## Statistical methods

Cox proportional hazard models were used to estimate HRs of mortality associated with PM and BC in cohort-specific analyses. Age was used as the underlying time variable for the baseline hazard. The regression model included adjustment for calendar year, baseline information on confounders as well as area-level socioeconomic. Since the Stockholm CEANS cohort consisted of four subcohorts that differed in age and time of recruitment, separate baseline hazards were used for the analyses within this cohort. We censored individuals at death by other causes, the end of the study period or time of permanent emigration from the study areas. Associations with PM and BC were assessed using two exposure windows; a moving average over 1–5 years preceding the event (lag1-5) and 6–10 years preceding the event (lag6-10). For inclusion, annual mean concentrations were required for at least 80% of the time window. These moving averages were used as time-varying exposures and therefore considers both between and within individual contrast during the study period. Besides these time windows, associations were also assessed with concentrations at the year of recruitment. HRs and 95% CIs were expressed per 10, 5 and 1 µg/m$^3$ for PM10, PM2.5 and BC, respectively, as well as per IQRs across the three study areas.

Meta-analyses of cohort-specific estimates were performed using random effects.[33] Heterogeneity between cohort estimates were assessed by the $I^2$ statistic and statistically tested for deviations from homogeneity by a $\chi^2$ test based on the Cochran's Q statistic.

## RESULTS
### Participant characteristics

The total number of individuals included in our meta-analysis was 68 679, with the number of individuals included in each cohort presented in table 1. Total number of deaths by natural causes was 7 344. Age at recruitment differed largely between cohorts with the highest median in the Gothenburg PPS cohort and lowest in VIP Umeå, 69 compared with 40 years. By design only males were recruited to the Gothenburg PPS cohort, whereas the other three cohorts had a slightly larger proportion of women compared with men. Baseline current smoking

was highest in the Gothenburg PPS cohort and lowest in Stockholm CEANS and VIP Umeå, 39% compared with about 20%. Leisure time physical activity had a different categorisation within the Stockholm CEANS cohort but it can be concluded that there was a larger proportion of individuals reporting a sedentary lifestyle at baseline in this cohort. The cohorts in Gothenburg had the largest proportion of physically active individuals, with higher frequency and intensity. Alcohol consumption was more frequent in Stockholm CEANS compared with VIP Umeå. University education was most common in Stockholm CEANS and VIP Umeå (above 30%) and less common in Gothenburg. A larger proportion of individuals were gainfully employed in the younger VIP Umeå than in Stockholm CEANS, 84% compared with 66%. Blue-collar occupation was more likely within the Gothenburg PPS cohort compared with Stockholm CEANS.

### Particle concentrations

Particle concentrations of PM10, PM2.5 and BC all had the highest mean and IQR in the Gothenburg cohorts and the lowest in VIP Umeå (figures 1–3). In total, means of lag1–5 PM10, PM2.5 and BC were; 11.3, 6.9 and 0.6 µg/m$^3$, respectively. Lag1–5 concentrations were on average lower than lag6–10 concentrations due to a decreasing trend in concentrations during the study period. The Pearson correlation coefficient between lag 1–5 concentrations of PM10 and PM2.5 was 0.88, PM10 and BC 0.71, and between PM2.5 and BC 0.75.

### Associations with mortality

Meta-estimates showed increased risks associated with PM10, PM2.5 and BC in relation to natural mortality; for lag1–5 17% (95% CI 6% to 31%) increased risk per 10 µg/m$^3$ PM10, 13% (95% CI -9% to 40%) increased risk per 5 µg/m$^3$ PM2.5 and 9% (95% CI 0% to 18%) increased risk per 1 µg/m$^3$ BC (figures 4–6; crude estimates in online supplemental material figure S1A–C). The statistical uncertainty was however considerable for the combined risk increase associated with PM2.5 and the CI included 1. For PM10 and PM2.5 higher risk estimates were found for lag1–5 concentrations compared with lag6–10, whereas for BC, the risk estimates were the same. Increased risks were estimated for all cohorts except for Gothenburg MONICA, where the HR estimate for natural mortality was below 1.

The risk estimates for CVD mortality associated with PM10 and PM2.5 appeared higher compared with natural mortality; 19% (95% CI 1% to 40%) and 23% (95% CI 3% to 48%) increased risks for lag1–5, per 10 and 5 µg/m$^3$, respectively (figures 7–9; crude estimates in online supplemental material figures S2A-B). The increased risks associated with BC seemed lower for CVD mortality than for natural mortality for lag1–5, but similar for lag6–10 (figure 9; crude estimates in online supplemental material figure S2C).

The HRs per IQR for natural mortality were higher for PM2.5 and PM10 than for BC; 6%, 5% and 2% increased

**Table 1** Descriptive statistics of the study participants in each cohort

| | Gothenburg, MONICA | Gothenburg, PPS | Stockholm, CEANS | Umeå, VIP |
|---|---|---|---|---|
| Participants (n) | 4 500 | 5 850 | 22 314 | 42 580 |
| Baseline data collection (calendar years) | 1985,1990,1995 | 1970–1974 | 1992–2004 | 1992–2014 |
| Age at enrolment (years; median (range)) | 46 (25–66) | 51 (47–56) | 56 (35–104) | 40 (40–50) |
| Women (%) | 52 | 0 | 58 | 52 |
| Smoking status (%) | | | | |
| Current smoker | 29 | 39 | 22 | 19 |
| Former smoker | N/A | 33 | 36 | 30 |
| Never smoker | 65 | 27 | 40 | 49 |
| Missing data | 5 | 0 | 1.8 | 1 |
| Physical activity (%) | | | | |
| Once a month or less/<1 hour/week | N/A | N/A | 61 | N/A |
| About once a month/~1 hour/week | N/A | N/A | 26 | N/A |
| three times a week or more/>2 hours/week | N/A | N/A | 7.6 | N/A |
| Missing data | N/A | N/A | 4.6 | N/A |
| Leisure time physical activity (%) | | | | |
| Sedentary | 18 | 24 | N/A | 36 |
| Moderate | 62 | 59 | N/A | 42 |
| Intermediate and vigorous | 18 | 17 | N/A | 22 |
| Missing data | 2 | 1 | N/A | 2 |
| Alcohol consumption (%) | | | | |
| Daily | N/A | N/A | 6.7 | 2 |
| Weekly | N/A | N/A | 55 | 16 |
| Seldom | N/A | N/A | 31 | 42 |
| Never | N/A | N/A | 5.6 | 1 |
| Missing data | N/A | N/A | 1.8 | 39 |
| Married/living with partner (%) | | | | |
| No | 21 | 14 | 29 | 23 |
| Yes | 47 | 86 | 70 | 77 |
| Missing data | 32 | <0.1 | 1.3 | 1 |
| Education level (%) | | | | |
| Primary school or less | 13 | N/A | 30 | 30 |
| Up to secondary school or equivalent | 32* | N/A | 36 | 30 |
| University degree and more | 20 | N/A | 31 | 40 |
| Missing data | 35 | N/A | 2.8 | 1 |
| Occupation (%) | | | | |
| Gainfully employed | N/A | N/A | 66 | 85 |
| Unemployed/not gainfully empolyed | N/A | N/A | 5.8 | 6 |
| Retired | N/A | N/A | 27 | 4 |
| Missing data | N/A | N/A | 1.3 | 4 |
| Socioeconomic index by occupation (%) | | | | |
| Blue collar | N/A | 48 | 27 | N/A |
| Low and intermediate white collar and self-employed | N/A | 23 | 51 | N/A |

**Table 1** Continued

|  | Gothenburg, MONICA | Gothenburg, PPS | Stockholm, CEANS | Umeå, VIP |
|---|---|---|---|---|
| High level white collar and self-employed professional with academic degrees | N/A | 30 | 18 | N/A |
| Missing data | N/A | 0 | 4.4 | N/A |
| Mean income by SAMS 1994 (SEK) | 154 780 | 148 602 | N/A | 130 000 |
| Mean income by SAMS 2009 (SEK) | N/A | N/A | 303 910 | N/A |

*Includes technical training.
CEANS, Cardiovascular Effects of Air Pollution and Noise Study; MONICA, Multinational Monitoring of Trends and Determinants in Cardiovascular Diseases; N/A, not available; PPS, Primary Prevention Study; SAMS, Small Areas for Market Statistics; SEK, Swedish Krona; VIP, Västerbotten Intervention Programme.

risk, respectively. The same order per IQR was found for CVD mortality; 10%, 5% and 1% for PM2.5, PM10 and BC, respectively.

No associations between PM and lung cancer or non-malignant respiratory mortality were observed (online supplemental material figures S3A-C and S4A–C). Increased risk was, however, observed for mortality by other causes (that is other than CVD, lung cancer, respiratory disease and external causes) with the same magnitude as observed for natural and CVD mortality (online supplemental material figures S5A–C).

For comparison with previous studies, associations were also assessed with residential PM concentrations for the recruitment year. These estimates were in general lower compared with those with lag1–5 and lag6–10 concentrations (online supplemental material figures S6A-B). Estimated risk increases for natural mortality were 7% (95% CI −4% to 18%), 8% (95% CI −2% to 20%) and 3% (95% CI −4% to 11%) in association with PM10, PM2.5 and BC, per 10, 5 and 1 µg/m³, respectively.

The heterogeneity between cohort HR estimates in relation to precision was in general very low. Heterogeneity was, however, observed for associations with lag1–5 PM2.5 and lag6–10 BC in relation to natural mortality, although non-of these were statistically significant (figures 4–9).

## DISCUSSION

We observed in four Swedish cohorts located in Gothenburg, Stockholm and Umeå increased risks associated with long-term residential outdoor levels of PM10, PM2.5 and BC in relation to natural and CVD mortality. PM10 was most consistently associated with natural and CVD mortality, whereas larger uncertainties were found for the associations between PM2.5 and CVD mortality and BC and natural mortality. Risk estimates associated with PM10 and PM2.5 appeared larger for exposures during the last 5 years in comparison to 6–10 years prior. For BC the associated risks

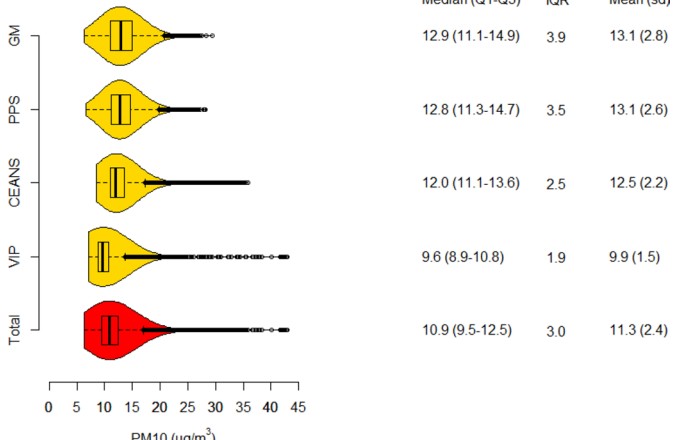

**Figure 1** Boxplot and density function of PM10 concentrations, in total and separate for each cohort. Tabulated values are quartile limits together with means and SD. CEANS, Cardiovascular Effects of Air Pollution and Noise Study; MONICA, Multinational Monitoring of Trends and Determinants in Cardiovascular Diseases; PM, particulate matter; PPS, Primary Prevention Study; VIP, Västerbotten Intervention Programme.

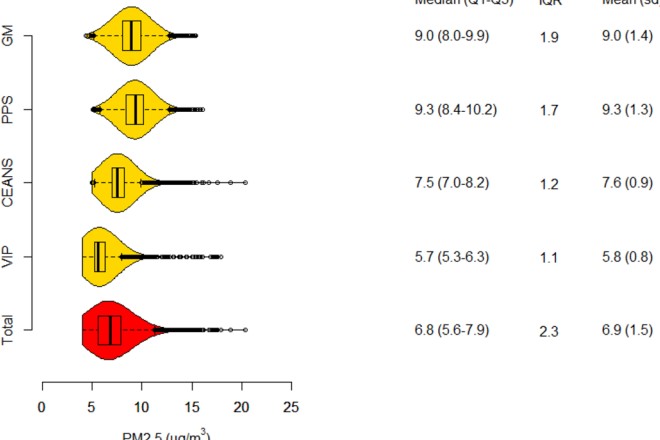

**Figure 2** Boxplot and density function of PM2.5 concentrations, in total and separate for each cohort. Tabulated values are quartile limits together with means and SD. CEANS, Cardiovascular Effects of Air Pollution and Noise Study; MONICA, Multinational Monitoring of Trends and Determinants in Cardiovascular Diseases; PM, particulate matter; PPS, Primary Prevention Study; VIP, Västerbotten Intervention Programme.

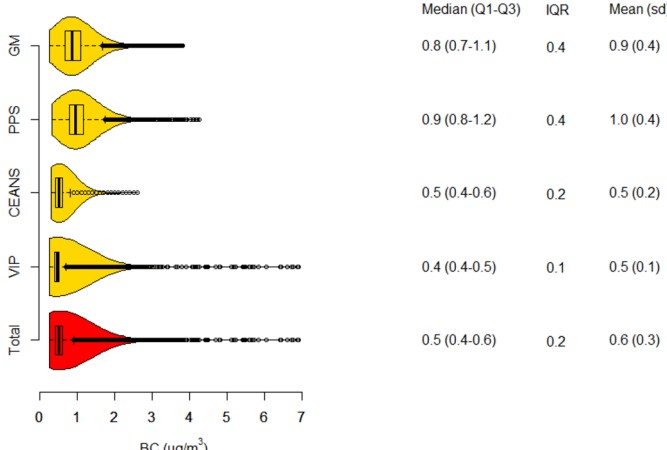

| | Median (Q1-Q3) | IQR | Mean (sd) |
|---|---|---|---|
| GM | 0.8 (0.7-1.1) | 0.4 | 0.9 (0.4) |
| PPS | 0.9 (0.8-1.2) | 0.4 | 1.0 (0.4) |
| CEANS | 0.5 (0.4-0.6) | 0.2 | 0.5 (0.2) |
| VIP | 0.4 (0.4-0.5) | 0.1 | 0.5 (0.1) |
| Total | 0.5 (0.4-0.6) | 0.2 | 0.6 (0.3) |

**Figure 3** Boxplot and density function of BC concentrations, in total and separate for each cohort. Tabulated values are quartile limits together with means and SD. BC, black carbon; CEANS, Cardiovascular Effects of Air Pollution and Noise Study; MONICA, Multinational Monitoring of Trends and Determinants in Cardiovascular Diseases; PPS, Primary Prevention Study; VIP, Västerbotten Intervention Programme.

were the same for both exposure windows in relation to natural mortality but higher for the exposure window 6–10 years prior in relation to CVD mortality. Within a previous meta-analysis of these cohorts, an association was found between BC and stroke incidence.[22] No association was, however, found between PM10 or PM2.5 and incidence in ischaemic heart disease.

The observed 17% increased risk per 10 $\mu g/m^3$ PM10 in relation to natural mortality is high compared with most studies (eg, the 6% increased risk found by Hoek et al[2] but similar to the recent findings within a Danish cohort presenting 12% increased risk (95% CI 3% to 22%) associated with a 15-year moving average of residential ambient concentrations.[6] Their estimate for CVD mortality was 30% (95% CI 11% to 53%) which was higher than our meta-estimate of 19%. For PM2.5 our observed risk increase by 13% per 5 $\mu g/m^3$ for natural mortality is also in agreement with the Danish estimate of 13% (95 CI 5% to 21%). As for PM10 the risk increase for CVD mortality associated with PM2.5 was again lower in our study, 23% compared with 29% (95% CI 13% to 47%). The average PM10 and PM2.5 exposure levels were 2.2 and 2.6 times higher in the Danish cohort compared with our study (lag1–10 time-weighted concentrations from the Danish cohort compared with lag1–5 moving averages in our study).

Using less detailed assessments of outdoor residential exposures, a meta-analysis of 22 cohorts within the ESCAPE project reported a borderline significant 4% (95% CI 0% to 9%) increased risk of natural-cause mortality for a 10 $\mu g/m^3$ difference in annual PM10 at the year of recruitment.[4] No association with PM10 was, however, observed for deaths in ischaemic heart disease or myocardial infarction, whereas a suggested association was found in relation to deaths in cerebrovascular

disease; 22% (95% CI –9% to 63%) increased risk per 10 $\mu g/m^3$.[5] The findings for mortality associated with PM2.5 have been heterogeneous, however, the meta-estimate among the ESCAPE cohorts showed a 7% (95% CI 2% to 13%) increased risk per 5 $\mu g/m^3$ of natural mortality and 21% (95% CI –13% to 69%) increased risk of death in cerebrovascular disease. As for PM10 no association was found in relation to death in ischaemic heart disease or myocardial infarction. Within earlier studies; a review and meta-analysis by Hoek et al found 6% (95% CI 4% to 8%) and 15% (95% CI 4% to 27%) increased risks of all-cause mortality and cardiovascular mortality per 5 $\mu g/m^3$, respectively. A stronger association between PM and mortality with cardiovascular causes compared with all natural mortality has also been reported by others,[1] and is in agreement with our findings on CVD mortality; and most apparent in relation to exposures during the last 5 years.

In accordance with our meta-estimate for BC the Danish cohort reported a risk increase by 9% (95% CI 4% to 15%) per 1 $\mu g/m3$ for all-cause mortality.[6] An increased risk by 16% (95% CI 5% to 27%) was also observed for CVD mortality, higher than the 5% increased risk found in our study. In the review by Hoek et al elemental carbon was found associated with a 6% (95% CI 5% to 7%) increased risk of all-cause mortality per 1 $\mu g/m^3$.

The lack of association between PM and non-malignant respiratory mortality in our study is in agreement with results from both the Danish study,[6] the ESCAPE study[7] and a large cohort study in Rome.[34] Previous studies have, however, found an increased risk of lung cancer incidence associated with both PM10 and PM2.5.[35 36] The statistical power for these outcomes was low in our meta-analyses due to the limited number of deaths in lung cancer and other respiratory causes.

The observed heterogeneity of PM associated risk for premature mortality between studies is possibly related to differences in particle composition, housing with differences in particle infiltration, population characteristics and the minor differences in confounding adjustments. The air pollution dispersion model used for the exposure assessment in our study has a high spatial resolution, and for the follow-up period it was also updated with new regional and local emission inventories as well as PM measurements to provide annual mean temporal PM concentrations. A higher precision in the exposure estimate have been found to produce higher relative health risk estimates.[13] The mortality risk associated with PM2.5 has recently been reported to be higher for local sources compared with distant sources, but with less precision in relative risk estimates, indicating the importance of high spatial resolution exposure data to successfully model differences in mortality.[37] The meta-analysis results in this study and the results presented by Hvidtfeldt et al both generally provided higher HR estimates compared with previous mortality studies. Both studies also used moving averages based on annual mean concentrations at the place of residence. Besides exposure data with

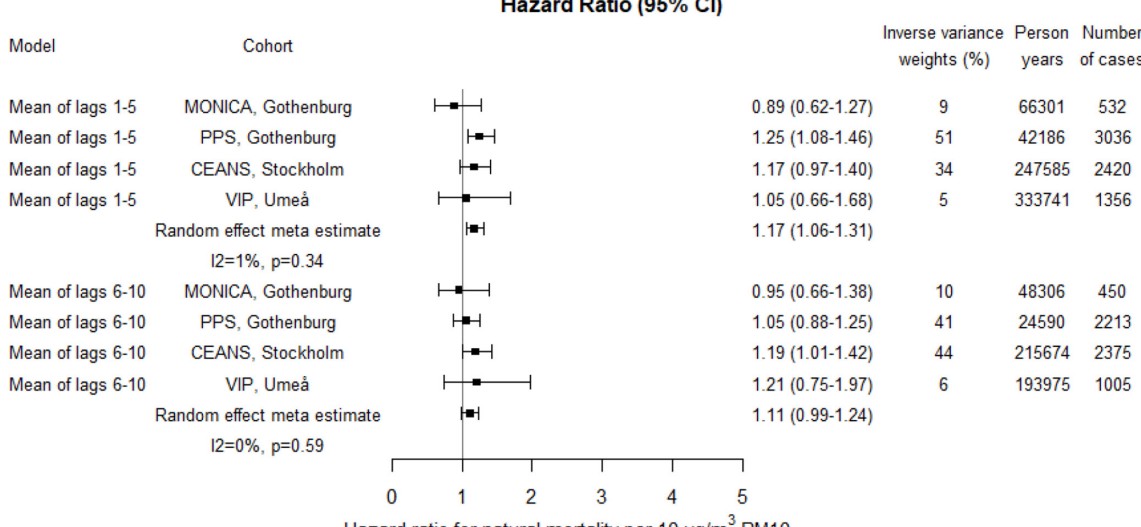

**Figure 4** Adjusted HRs for natural mortality associated with PM10 within each cohort and by random effect meta-analyses estimates. Exposures were assessed by moving average residential concentrations within the exposure windows last 5 years and 6–10 years prior. CEANS, Cardiovascular Effects of Air pollution and Noise Study; MONICA, Multinational Monitoring of Trends and Determinants in Cardiovascular Diseases; PM, particulate matter; PPS, Primary Prevention Study; VIP, Västerbotten Intervention Programme.

a high precision such a moving average over relevant time windows may be advantageous compared with residential concentrations at the year of recruitment or a time-weighted average exposure during the follow-up. These two studies also adjusted both for individual data on confounders and neighbourhood-level data on socioeconomics.

### Strengths and limitations

The strength of this study includes the state-of-the-art air pollution modelling used for exposure assessment. Several factors that affect the local dispersion of air pollution particles were considered including meteorological conditions, amount of traffic including vehicle types and speed, which affect exhaust and wear emissions, the width of the street and the height of neighbouring building. This facilitates models with a high spatial resolution providing the ability to estimate an individual's exposure at the residential address. Temporal differences for each address were also accounted for by repeatedly updating the exposure model during the period of follow-up. Our modelled concentrations have showed a good agreement when validated against measured concentrations by monitoring stations in city centres.[24] Due to uncertainties in data input to the dispersion model,

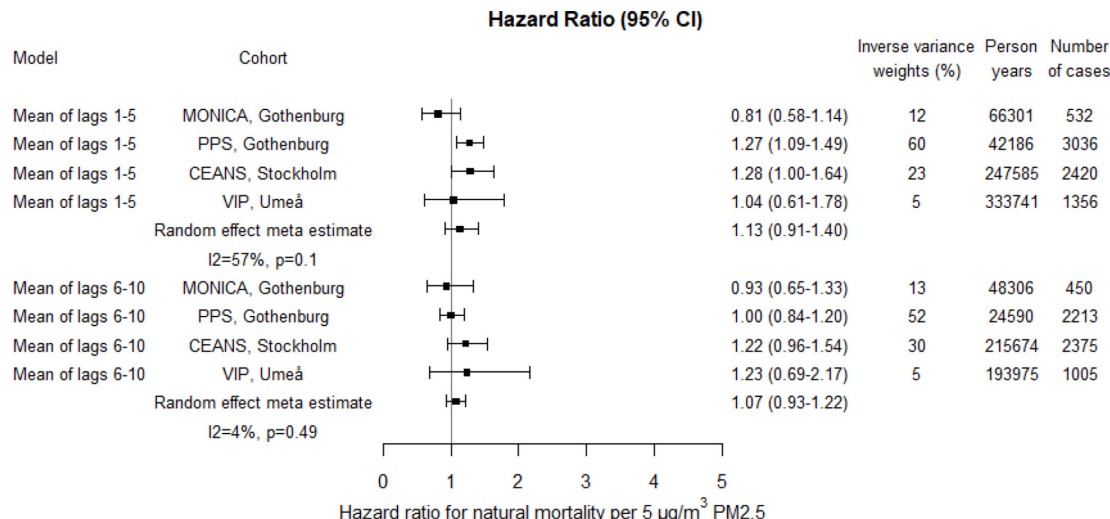

**Figure 5** Adjusted HRs for natural mortality associated with PM2.5 within each cohort and by random effect meta-analyses estimates. Exposures were assessed by moving average residential concentrations within the exposure windows last 5 years and 6–10 years prior. CEANS, Cardiovascular Effects of Air pollution and Noise Study; MONICA, Multinational Monitoring of Trends and Determinants in Cardiovascular Diseases; PM, particulate matter; PPS, Primary Prevention Study; VIP, Västerbotten Intervention Programme.

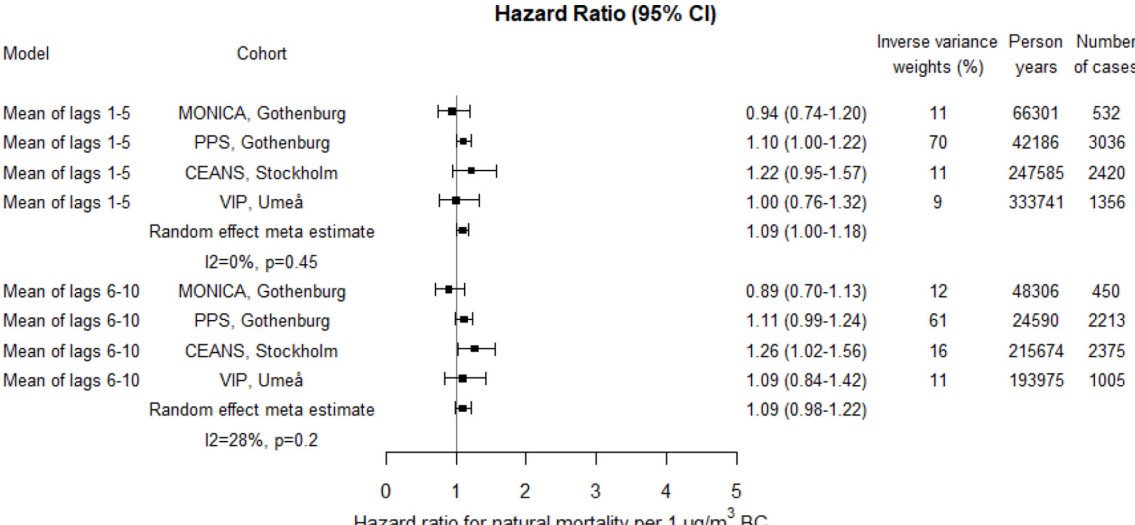

**Figure 6** Adjusted HRs for natural mortality associated with black carbon (BC) within each cohort and by random effect meta-analyses estimates. Exposures were assessed by moving average residential concentrations within the exposure windows last 5 years and 6–10 years prior. CEANS, Cardiovascular Effects of Air pollution and Noise Study; MONICA, Multinational Monitoring of Trends and Determinants in Cardiovascular Diseases; PM, particulate matter; PPS, Primary Prevention Study; VIP, Västerbotten Intervention Programme.

and since modelled PM levels at the home address do not equal true personal exposure, exposure misclassifications at the individual level will nevertheless arise. True personal exposure would for instance include indoor particle concentrations at the place of residence, at the workplace and during commuting. During the years of follow-up, there has been a decreasing trend in both total concentrations of PM and age-specific mortality. It was therefore important to adjust for calendar year in the Cox models.

Air pollution gases were not part of our hypothesis and thus not modelled. Studies using nitrogen dioxide (NO2) and nitrogen oxides as indicators of traffic-related air pollution have also reported associations with increased risk of premature mortality.[2 38] Even though there could be an NO2 associated short-term effect on mortality independent of particle matter (eg, Mills et al,[39] the evidence for long-term effects when controlling for traffic related PM is weaker and exposure to particles is considered as the primary causal determinant of health effects.[40]

As presented above the incidence in ischaemic heart disease and stroke have previously been assessed in relation to particle concentrations on this material.[22]

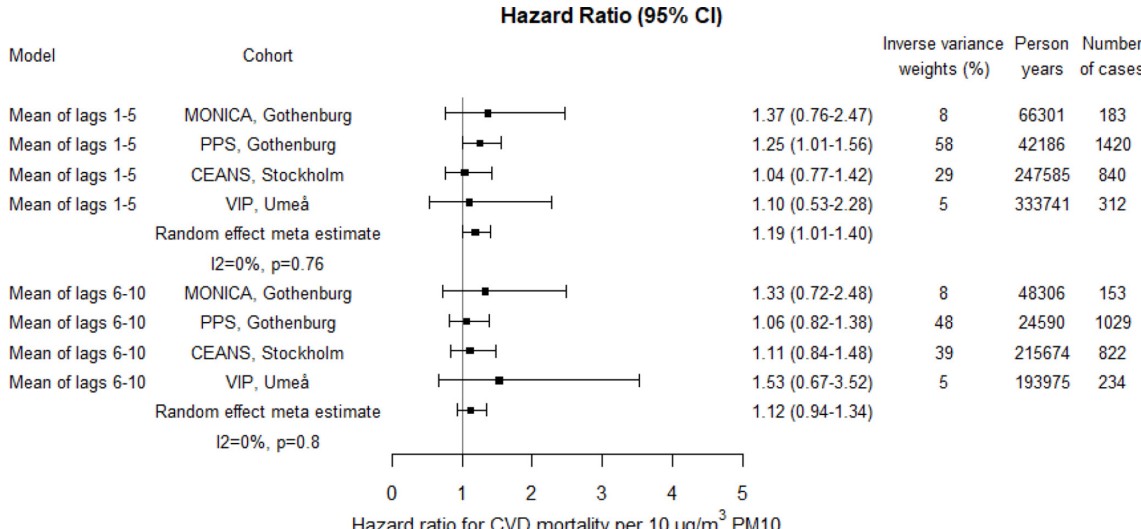

**Figure 7** Adjusted HRs for cardiovascular disease (CVD) mortality associated with PM10 within each cohort and by random effect meta-analyses estimates. Exposures were assessed by moving average residential concentrations within the exposure windows last 5 years and 6–10 years prior. CEANS, Cardiovascular Effects of Air pollution and Noise Study; MONICA, Multinational Monitoring of Trends and Determinants in Cardiovascular Diseases; PM, particulate matter; PPS, Primary Prevention Study; VIP, Västerbotten Intervention Programme.

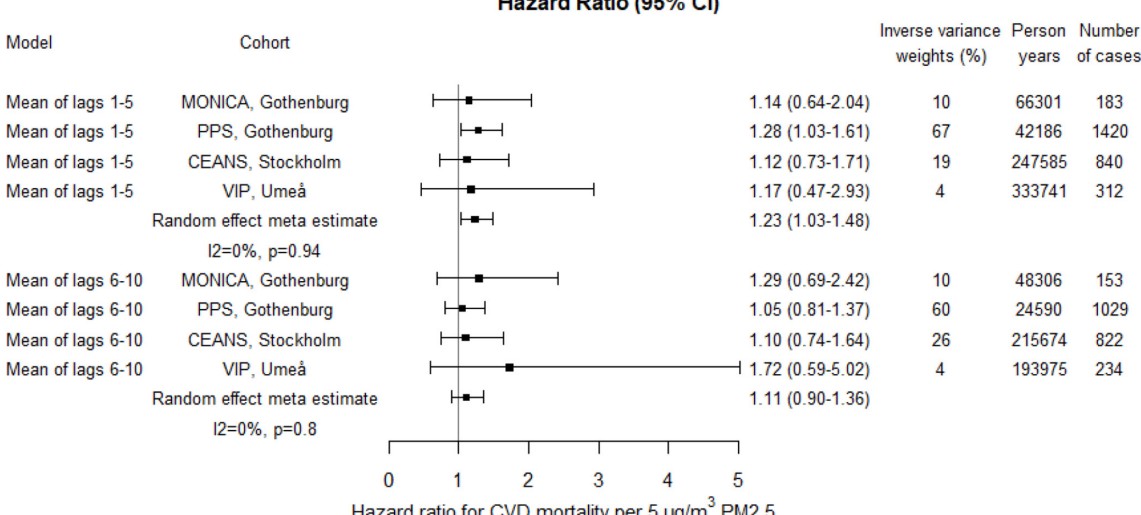

**Figure 8** Adjusted HRs for cardiovascular disease (CVD) mortality associated with PM2.5 within each cohort and by random effect meta-analyses estimates. Exposures were assessed by moving average residential concentrations within the exposure windows last 5 years and 6–10 years prior. CEANS, Cardiovascular Effects of Air pollution and Noise Study; MONICA, Multinational Monitoring of Trends and Determinants in Cardiovascular Diseases; PM, particulate matter; PPS, Primary Prevention Study; VIP, Västerbotten Intervention Programme.

If the same hypothesis is repeatedly tested on the same material then multiple testing would be an issue, affecting the confidence level of the uncertainty intervals presented for the estimated HRs. On the other hand, results with improvements in exposure data and aggregation with similar cohorts are important to present. Parts of the Stockholm cohorts used for this study were also included in the ESCAPE project estimating the long-term effects on human health of exposure to air pollution in Europe. The period of follow-up differs between these studies, and the exposure modelling and assessment, which were in the

ESCAPE-studies only performed for the year of inclusion and built on land use regression (LUR) models. Also one of the cohorts from Gothenburg (PPS) has been used to study effects of air pollution but with other exposure measures[41] and outcomes.[42]

Since this is an observational study it was also necessary to adjust for a set of potential confounders, which were prespecified. Mortality outcomes, measures of exposures (PM10, PM2.5 and BC) and time windows (time dependent moving averages over lags 1–5 and 6–10, respectively) were also a priori selected. Besides natural mortality associations were assessed with CVD

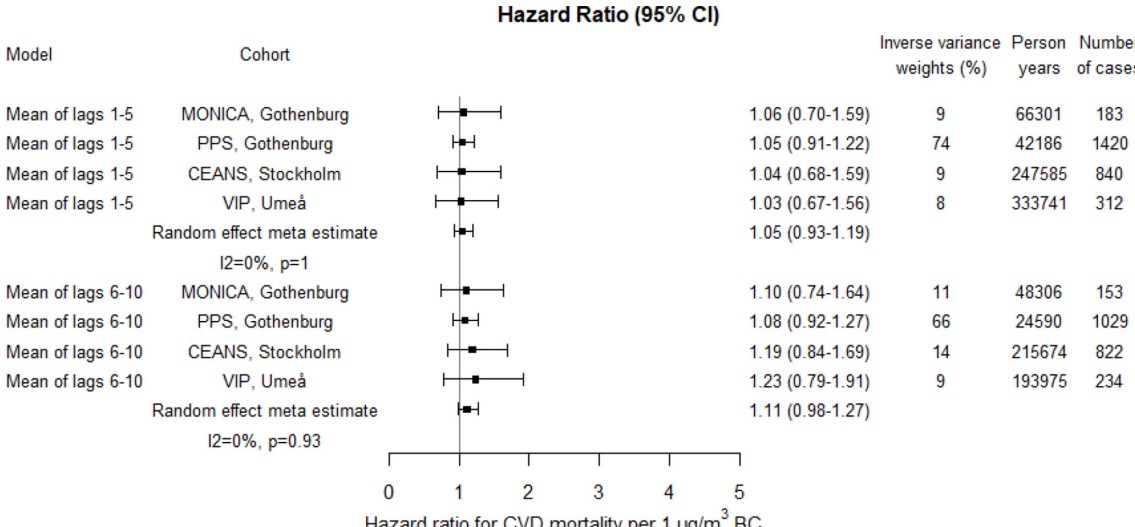

**Figure 9** Adjusted HRs for cardiovascular disease (CVD) mortality associated with BC within each cohort and by random effect meta-analyses estimates. Exposures were assessed by moving average residential concentrations within the exposure windows last 5 years and 6–10 years prior. BC, black carbon; CEANS, Cardiovascular Effects of Air pollution and Noise Study; MONICA, Multinational Monitoring of Trends and Determinants in Cardiovascular Diseases; PM, particulate matter; PPS, Primary Prevention Study; VIP, Västerbotten Intervention Programme.

and three (more) specific causes of death. A limitation that needs to be considered for more exploratory outcomes is that CIs were not constructed to maintain a 95% familywise error rate. For the interpretation of the results this means that even though the study does not repeatedly test the same hypothesis, for instance, in subgroups of participants, it is for cause-specific mortality results necessary to consider that the evidence of air pollution effects need to be judged together with other studies. Specifically, for mortality by other causes (other than CVD, lung cancer, respiratory disease and external causes), which has rarely been assessed in relation to air pollution, additional studies are needed. Limitations also include the lack of information on noise exposure and green space near the home address which could be confounders or effect-modifiers of air pollution effects.

Another strength of this study is the large number of cohort participants and ability to adjust for a large set of confounders at both individual and area level, however, information on confounders from questionnaires was only available at baseline. Even though we expect the cohort participants with age ranging from 40 years to have fairly stable lifestyles and habits, such baseline variables may change, for example, due to disease. No clear heterogeneity in associations between cohorts was observed, however, with only four cohorts the test had low statistical power to reject the null hypothesis of homogeneity.

This study additionally contributes with HR estimates between PM and mortality in the lower range of exposure with PM2.5 ranging between 4.0 and $20.4\,\mu g/m^3$ and PM10 ranging between 6.3 and $41.9\,\mu g/m^3$. Even though this is lower than the current EU standards of $25\,\mu g/m^3$ for PM2.5 and (with a few exceptions) lower than $40\,\mu g/m^3$ for PM10, the study found higher relative risks for premature mortality compared with most previous studies even though mean exposures in those studies have been higher.

## Conclusions

This study of four cohorts with relatively low exposures showed increased risks of mortality in association with long-term exposure to PM10, PM2.5 and BC, and thus strongly support long-term air pollution associated effects. Since high-risk increases were observed even at relatively low exposures, the findings are relevant for future decisions concerning air quality policies.

## Author affiliations
[1]Section of Sustainable Health, Department of Public Health and Clinical Medicine, Umea University, Umeå, Sweden
[2]Occupational and Environmental Medicine, Department of Public Health and Community Medicine, Institute of medicine, Sahlgrenska Academy, University of Gothenburg & Sahlgrenska University Hospital, Gothenburg, Sweden
[3]Institute of Environmental Medicine, Karolinska Institutet, Stockholm, Sweden
[4]Department of Cardiology, Danderyd Hospital, Stockholm, Sweden
[5]Swedish Meteorological and Hydrological Institute, Norrkoping, Sweden
[6]SLB-analys, Environment and Health Administration, Stockholm, Sweden
[7]Family Medicine, Public Health and Clinical Medicine, Umeå University, Umeå, Sweden
[8]Department of Molecular and Clinical Medicine, Institute of Medicine, University of Gothenburg, Sahlgrenska University Hospital, Goteborg, Sweden
[9]Ageing Research Center, Department of Neurobiology, Care Sciences and Society, Karolinska Institutet and Stockholm University, Stockholm, Sweden
[10]Stockholm Gerontology Research Center, Stockholm, Sweden
[11]Centre for Epidemiology and Community Medicine, Stockholm County Council, Stockholm, Sweden
[12]Department of Public Health Science, Karolinska Institutet, Stockholm, Sweden
[13]Department of Medical Epidemiology and Biostatistics, Karolinska Institutet, Stockholm, Sweden
[14]Department of Environmental Science, Stockholm University, Stockholm, Sweden
[15]Centre for Occupational and Environmental Medicine, Region Stockholm, Stockholm, Sweden

**Contributors** JS performed the meta-analysis and wrote with support from BF the first draft manuscript. JS, EMA, NA and GS conducted the cohort analyses. DS, KE, LG and CJ were responsible for dispersion modelling of particles. PW, AR, DR, KL, AL and PKEM and were responsible for collection of cohort data. GP, BF, LB, TB, JS, GS, LS, PL, EMA and NA designed the epidemiological study. All authors contributed to the interpretation of the results, finalisation of the manuscript and approved the final manuscript.

**Funding** This study was funded by the Swedish Environmental Protection Agency as part of the Swedish Clean Air and Climate Research Program (grant NV-06576-13). BF and JS was also supported by NordForsk grant no. 75007. The GOT-MONICA and PPS cohorts were supported by the Swedish Research Council, 2013-5187 (SIMSAM). The SDPP cohort was funded by the Stockholm County Council, the Swedish Research Council, the Swedish Diabetes Association, the Novo Nordisk Scandinavia, and GlaxoSmithKline. The 60YO cohort was funded by the Stockholm County Council and the Swedish Research Council. The Swedish Twin Registry is managed by Karolinska Institutet and receives funding through the Swedish Research Council under the grant no. 2017-00641. Data collection in SALT was supported by a grant from the National Institutes of Health (NIH), grant no. 1R01 AG08724. The VIP-Umeå cohort was funded by the Västerbotten County Council. PL was supported by funding from the Swedish Research Council for Health, Working Life and Welfare (FORTE) 2015-00917 and Karolinska Institute's Strategic Research Area in Epidemiology (SFO-EPI). LB was financed by grants from the Swedish state under the agreement between the Swedish government and the county councils: the ALF-agreement (74580).

**Competing interests** None declared.

**Patient consent for publication** Not required.

**Ethics approval** The study was reviewed and approved by the Ethical Review Boards of Gothenburg (references T800-08 and T547-13), Stockholm (reference 2018/2064-32), and Umeå (reference 2015/16-31Ö).

**Provenance and peer review** Not commissioned; externally peer reviewed.

**Data availability statement** No data are available. No patient involved. The public was also not involved in the design, or conduct, or reporting, or dissemination plans of the research.

## ORCID iD
Johan Nilsson Sommar http://orcid.org/0000-0002-8854-498X

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
