## [Reviewer comments · BMJ Open]

ARTICLE DETAILS

TITLE (PROVISIONAL)	Long-term exposure to particulate air pollution and black carbon in relation to natural and cause specific mortality: a multi-cohort study in Sweden
AUTHORS	Sommar, Johan; Andersson, Eva M.; Andersson, Niklas; Sallsten, Gerd; Stockfelt, Leonard; Ljungman, Petter; Segersson, David; Eneroth, Kristina; Gidhagen, Lars; Molnar, Peter; Wennberg, Patrik; Rosengren, Annika; Rizzuto, Debora; Leander, Karin; Lager, Anton; Magnusson, Patrik; Johansson, Christer; Barregard, Lars; Bellander, Tom; Pershagen, Göran; Forsberg, Bertil

VERSION 1 – REVIEW

REVIEWER	Whanhee Lee Yale University
REVIEW RETURNED	08-Jan-2021

GENERAL COMMENTS	Thank you for giving me a chance to review this excellent paper. It was a big pleasure. This manuscript is quite solid and sophisticated; however, there are some points I would like to ask the authors to clarify or modify. Please consider them. Major comments. 1) Introduction: the authors applied different units for each pollutant (PM10: 10 µg/m³, BC: 1 µg/m³, PM2.5: 5 µg/m³) as the main results. It looks confusing, a little bit. Is there any reason why the authors did not consider IQR increase results as the main results? I think RRs per IQR increase are more comparable and powerful. 2) Introduction, Lines 9-13: As far as I understood, this study did not consider NO₂ and NO_x. This paragraph can give confusion to readers. Please consider whether this paragraph is necessary. 3) Introduction, Lines 19-20: What is the exact meaning of this sentence?: "The observed risk ~ not reach statistical significance". This sentence looks in conflict with the previous sentence (Lines 18-19). Please clarify it. 4) Statistical methods, Line 29-42: First, I'd like to ask reasons why this study applied "moving average" exposure rather than "time-varying" exposure. Moving average exposure is inherently limited in selecting proper lag periods (i.e. uncertainty of lag period selections). In addition, the moving average exposure method is highly limited in considering temporal changes of exposure and their effects on outcomes. I think the cohort data and setting used in this
---

	study can consider the time-varying Cox-proportional hazard models, which can consider “between” and “within (time-varying)” effects of patients at the same time. 5) Statistical methods, Line 43-45: The authors used the meta-analysis. Could I ask the reason why the authors performed meta-analyses? I surmise that they can calculate average estimates and each-cohort estimates simultaneously if they include “fixed or random” cohort effects in the Cox-proportional hazard models. R package “coxme” provides the random effect Cox model. This method can provide more statistically powerful estimates. Of course, I don’t want the authors to change their methods. The current method looks sufficiently appropriate. I just want to ask the reason. 6) Overall parts: As a statistician, I fully agree with the importance of “statistical significance”. However, it is very difficult to consent that the associations with >0.05 p-value do not have important meanings. This manuscript repeatedly described “non-reached statistical significance”. I think this sentence can bring misunderstands to readers. In addition, in statistical points of view, the repeated mentions about the statistical significance seem to derive the necessity of multiple comparison tests. I’d like to suggest reducing or deleting the mentions regarding statistical significance unless there are essential reasons. The current results look enough to be described as the main results. Minor comments: 1) Strengths and limitation of this study (the last sentence): the term "high" seems ambiguous. Compared to what? The authors should clarify the comparison group if they would like to use this term. If it means "positive", please revise this term with a more appropriate word. 2) Line 29: From Stockholm County, ?
--	--

REVIEWER	Stanley S. Young CGSTAT
REVIEW RETURNED	19-Jan-2021

GENERAL COMMENTS	bmjopen-2020-046040 To editor: Watermark preclude lifting text and tables for more careful examination. The meta-analysis results, Figure2a-c and following, are given as images. It would be useful to more technical readers to give these figures so the data can be copy/pasted into analysis programs. General comments This study combines for cohort studies covering three Swedish cities. The researchers come from the point of view that air quality is a health hazard based on previous literature, primarily time series studies. They add that meta-analysis of cohort studies can be used to examine the long-term consequence of poor air quality. They look at three air quality measures, PM10, PM2.5, and black carbon. They look at multiple health measures, Natural mortality (all-
--

cause), respiratory disease, cardiovascular disease, and lung cancer. They note that they have access to a full range of causes of deaths. They examine time windows of 1-5 years, 6-10 years, and a lag of 6-10 years. For exposure they use a mathematical dispersion model. A lot of questions are examined, air quality 3; health measures 4+; time periods 3+; demographic covariates, 8+. So, there are at least $2 \times 4 \times 3 \times 8 = 192$ questions/models at issue. All testing is done without any adjustment for multiple testing or multiple modeling, MTMM.

None of the literature cited to help evaluate/support their claims does anything to address multiple testing or multiple modeling. Two of the cohort studies used in this report are part of the larger studies, Primary Prevention Study and MONICA. A google scholar search of "MONICA project" returned just over one million hits.

As is common in this subject area, air pollution environmental epidemiology, few negative studies are cited in positive papers. Several negative studies come to mind.

Chay K, Dobkin C, Greenstone M. 2003. The Clean Air Act of 1970 and adult mortality. *Journal of Risk and Uncertainty* 27:279–300.

Enstrom JE. 2005. Fine particulate air pollution and total mortality among elderly Californians, 1973–2002. *Inhalation Toxicology* 17:803–816.

Milojevic, A., Wilkinson, P., Armstrong, B., Bhaskaran, K., Smeeth, L., Hajat, S., 2014. Short-term effects of air pollution on a range of cardiovascular events in England and Wales: case-crossover analysis of the MINAP database, hospital admissions and mortality. *Heart* 100:1093-1098.

Enstrom JE. 2017. Fine particulate matter and total mortality in Cancer Prevention Study cohort reanalysis. *Dose-Response: An International Journal*. 2017:1-12.

Young SS, Smith RL, Lopiano KK. (2017) Air quality and acute deaths in California, 2000-2012. *Regulatory Toxicology and Pharmacology* 88, 173-184.

It is not an open and shut case that the air quality/health effect claims are real.

A recent study funded by WHO gathered and examined many air quality/health effect papers. They used 27 papers to evaluate all-cause mortality. They give a meta-analysis risk ratio of 1.0065, which I consider small enough to call claims into question.

Orellano P, Reynoso J, Quaranta N, Bardach A, Ciapponi A. 2020. Short-term exposure to particulate matter (PM10 and PM2.5), nitrogen dioxide (NO2), and ozone (O3) and all-cause and cause-specific mortality: Systematic review and meta-analysis. *Environment International*. 142, 105676. <https://doi.org/10.1016/j.envint.2020.105876>

The authors have done a lot of work. Is there a way forward? The authors could use their results, Figure 2a-c, Figure 3a-c, figure S1a-c, Figure S2a-c, etc., to present a much better statistical picture of their results. Each estimated risk ratio and confidence limits can be used to compute a p-value. If you order the p-values from smallest to largest, they can be plotted against the integers, 1, 2, 3, ... to give a p-value plot. The reader can see all the statistical tests in the context of the other tests in one figure. Also, the p-values can be

	used to produce multiplicity adjusted p-values so that the reader can judge the statistical reliability of the individual p-values/claims. Supplemental Information for meta-analysis evaluation S. Stanley Young, CGStat, Raleigh NC 27607 Warren Kindzierski, School of Public Health, University of Alberta, Edmonton, Alberta T6G 1C9. See arXiv 1808.04408-1. I think this addition to the paper would help the reader evaluate the work. If the authors choose to go in this direction, I am willing to help. Specific comments Abstract, page 2, line 31 In the abstract the authors say, "These findings are relevant for future decisions concerning air quality policies." Two points: Until this research area deals adequately with multiple testing and multiple modeling, research claims are unreliable. Recently the US EPA, in the interest of research transparency, want public access to data used in key papers used for regulatory decisions. Page 4, line 3/6 Enstrom 2017 gives a meta-anlysis of cohort studies and finds no effect. Young and Kindzierski 2019 examine a meta-analysis that appeared in JAMA and find no mortality effects on heart attacks. Unless authors deal effectively with multiple testing and multiple modeling their claims are unreliable for serious regulatory decisions. Young SS, Kindzierski KB. 2019. Evaluation of a meta-analysis of air quality and heart attacks, a case study, Critical Reviews in Toxicology, doi: 10.1080/10408444.2019.1576587 Page 6, line 15/10 The authors examine PM10, PM2.5, and black carbon. They should note if other air components were available and were they examined? Page 6, line 28/20 It is common to use statistical significance of 0.05 with no correction for multiple testing or multiple modeling in environmental epidemiology. Until there is better statistical control of false positives, authors and readers should keep in mind the statistical reliability of these studies. The authors should comment. Young SS. (2017) Air quality environmental epidemiology studies are unreliable. Regulatory Toxicology and Pharmacology 88, 177-180. Page 6, line 38/28 The predictor variables, PM10, PM2.5, and black carbon, are highly correlated among themselves and likely over time. The authors should provide somewhere a correlation matrix. Page 6, line 41/30 The words ESCAPE project appear in 93 titles of journal papers. It appears that data from this project has been used to address many questions. Page 7, line 9/5 This paper is dependent on the reliability of the dispersion model, among many other things. I'm not familiar with this model. Page 8, line 12/8. I suspect that Ethical Review Boards are not familiar with multiple testing/multiple modeling, so they do not raise the MTMM issue with researchers.
--	---

	Page 8, line25/17ff. The authors note that they had access to many other health effects. Page 9, line 20/14. Confounding variables present a problem for observational studies. Yes, researchers would like to statistically remove possible biases and yes, they can be used to decrease the experimental error, BUT they can lead to try this and try that until an effect is found. And, important confounding variables might not be included. For example, radon is an important confounder for lung cancer. See N Engl J Med 1994; 330:159-164 DOI: 10.1056/NEJM199401203300302 “The estimates of risk were in the same range as those projected from data in miners. The interaction between radon exposure and smoking with regard to lung cancer exceeded additivity and was closer to a multiplicative effect.” Page 10, line 31/22 These results suggest that low dose results are largely from Umea. Location and dose appear to be confounded. Page 11, lines 28/20 The health effects of stroke, ischemic heart disease are mentioned. It would help the reader to know about claims made using one or more of the cohorts to better get an idea of the extent of multiple testing and multiple modeling. Page 12, lines 11/7 It seems clear to me that none of the cited papers adjusted for multiple testing or multiple modeling. Page 12, lines 20/14 None of the predictor variables, PM10, PM2.5, and black carbon, are chemically defined, so PM2.5 in one location might be radically different from PM2.5 in another locations. Page 13, Lines 22/15ff. Conclusions. With a p-value plot and multiplicity adjusted p-values these conclusions should be reexamined.
--	---

VERSION 1 – AUTHOR RESPONSE

Reviewer 1.

Comments to the Author:

Thank you for giving me a chance to review this excellent paper. It was a big pleasure. This manuscript is quite solid and sophisticated; however, there are some points I would like to ask the authors to clarify or modify. Please consider them.

Major comments.

R1.1. Introduction: the authors applied different units for each pollutant (PM10: 10 µg/m3, BC: 1 µg/m3, PM2.5: 5 µg/m3) as the main results. It looks confusing, a little bit. Is there any reason why the authors did not consider IQR increase results as the main results? I think RRs per IQR increase are more comparable and powerful.

***Response:** The increments chosen are to facilitate comparisons with previous studies. Reporting hazard ratios per inter-quartile range would focus on comparing hazard ratios between pollutants. Even though this was not the main aim of this study such a comparison was presented under results (page 9, lines 3-5).

R1.2. Introduction, Lines 9-13: As far as I understood, this study did not consider NO2 and NOx. This paragraph can give confusion to readers. Please consider whether this paragraph is necessary.

***Response:** We understand and have excluded the paragraph (page 4, lines 9-13).

R1.3. Introduction, Lines 19-20: What is the exact meaning of this sentence?: "The observed risk ~ not reach statistical significance". This sentence looks in conflict with the previous sentence (Lines 18-19). Please clarify it.

***Response:** Corrected (page 4, lines 19-20), we thank the reviewer for identifying the mistake.

R1.4. Statistical methods, Line 29-42: First, I'd like to ask reasons why this study applied "moving average" exposure rather than "time-varying" exposure. Moving average exposure is inherently limited in selecting proper lag periods (i.e. uncertainty of lag period selections). In addition, the moving average exposure method is highly limited in considering temporal changes of exposure and their effects on outcomes. I think the cohort data and setting used in this study can consider the time-varying Cox-proportional hazard models, which can consider "between" and "within (time-varying)" effects of patients at the same time.

***Response:** The study did use time-varying exposures. We realize that this was not clear and a clarification of this has been added (page 7, lines 41-43).

R1.5. Statistical methods, Line 43-45: The authors used the meta-analysis. Could I ask the reason why the authors performed meta-analyses? I surmise that they can calculate average estimates and each-cohort estimates simultaneously if they include "fixed or random" cohort effects in the Cox-proportional hazard models. R package "coxme" provides the random effect Cox model. This method can provide more statistically powerful estimates. Of course, I don't want the authors to change their methods. The current method looks sufficiently appropriate. I just want to ask the reason.

***Response:** This was chosen by necessity since the ethical permissions did not allow for merging of the data.

R1.6. Overall parts: As a statistician, I fully agree with the importance of "statistical significance". However, it is very difficult to consent that the associations with >0.05 p-value do not have important meanings. This manuscript repeatedly described "non-reached statistical significance". I think this sentence can bring misunderstands to readers. In addition, in statistical points of view, the repeated mentions about the statistical significance seem to derive the necessity of multiple comparison tests. I'd like to suggest reducing or deleting the mentions regarding statistical significance unless there are essential reasons. The current results look enough to be described as the main results.

***Response:** We agree with the reviewer and have made corresponding changes in terms of statistical significance (page 8, lines 34-36; page 9, lines 9-10, 18-19, and 24-26).

Minor comments:

R1.7. Strengths and limitation of this study (the last sentence): the term "high" seems ambiguous. Compared to what? The authors should clarify the comparison group if they would like to use this term. If it means "positive", please revise this term with a more appropriate word.

***Response:** We agree and have now clarified that this compared with previous studies (page 11, lines 25-27).

R1.8. Line 29: From Stockholm County, ?

***Response:** A clarification has been made (page 5, line 29).

Review 2.

bmjopen-2020-046040

To editor: Watermark preclude lifting text and tables for more careful examination.

The meta-analysis results, Figure2a-c and following, are given as images. It would be useful to more technical readers to give these figures so the data can be copy/pasted into analysis programs.

General comments

This study combines for cohort studies covering three Swedish cities. The researchers come from the point of view that air quality is a health hazard based on previous literature, primarily time series studies. They add that meta-analysis of cohort studies can be used to examine the long- term consequence of poor air quality.

R2.1. They look at three air quality measures, PM10, PM2.5, and black carbon. They look at multiple health measures, Natural mortality (all-cause), respiratory disease, cardiovascular disease, and lung cancer. They note that they have access to a full ranged of causes of deaths. They examine time windows of 1-5 years, 6-10 years, and a lag of 6-10 years. For exposure they mathematical dispersion model. A lot of questions are examined, air quality 3; health measures 4+; time periods 3+; demographic covariates, 8+. So, there are at least $2 \times 4 \times 3 \times 8 = 192$ questions/models at issue. All testing is done without any adjustment for multiple testing or multiple modeling, MTMM.

*Response: The authors would like to note that this is not the number of hypotheses under study nor the number of models fitted but rather the total number of parameters estimated.

R2.2. None of the literature cited to help evaluate/support their claims does anything to address multiple testing or multiple modeling. Two of the cohort studies used in this report are part of the larger studies, Primary Prevention Study and MONICA. A google scholar search of "MONICA project" returned just over one million hits.

As is common in this subject area, air pollution environmental epidemiology, few negative studies are cited in positive papers. Several negative studies come to mind.

Chay K, Dobkin C, Greenstone M. 2003. The Clean Air Act of 1970 and adult mortality. *Journal of Risk and Uncertainty* 27:279–300.

Enstrom JE. 2005. Fine particulate air pollution and total mortality among elderly Californians, 1973–2002. *Inhalation Toxicology* 17:803–816.

Milojevic, A., Wilkinson, P., Armstrong, B., Bhaskaran, K., Smeeth, L., Hajat, S., 2014. Short-term effects of air pollution on a range of cardiovascular events in England and Wales: case-crossover analysis of the MINAP database, hospital admissions and mortality. *Heart* 100:1093-1098.

Enstrom JE. 2017. Fine particulate matter and total mortality in Cancer Prevention Study cohort reanalysis. *Dose-Response: An International Journal*. 2017:1-12.

Young SS, Smith RL, Lopiano KK. (2017) Air quality and acute deaths in California, 2000- 2012. *Regulatory Toxicology and Pharmacology* 88, 173-184.

It is not an open and shut case that the air quality/health effect claims are real.

*Response: These studies were not cited since they either estimate short-term effects or have methodological flaws that may to a large extent have biased the results. This critique has also been published (Brunekreef and Hoek, 2006; Gapstur and Brawley, 2017; Pope et al, 2017).

Referenses:

Brunekreef B, Hoek G. A critique of "fine particulate air pollution and total mortality among elderly Californians, 1973-2002" by James E. Enstrom. *Inhal Toxicol*. 2006 Jun;18(7):507-8; discussuin 509-14. doi: 10.1080/08958370600596219. PMID: 16603482.

Gapstur SM, Brawley OW. Re: Fine Particulate Matter and Total Mortality in Cancer Prevention Study Cohort Reanalysis. *Dose-Response*. October 2017. doi:10.1177/1559325817749412

Pope CA, 3rd, Krewski D, Gapstur SM, et al. Fine Particulate Air Pollution and Mortality: Response to Enstrom's Reanalysis of the American Cancer Society Cancer Prevention Study II Cohort. *Dose Response* 2017;15(4):1559325817746303. doi: 10.1177/1559325817746303 [published Online First: 2017/12/26]

R2.3. A recent study funded by WHO gathered and examine many air quality/health effect papers. They used 27 papers to evaluate all-cause mortality. They give a meta-analysis risk ratio of 1.0065, which I consider small enough to call claims into question.

Orellano P, Reynoso J, Quaranta N, Bardach A, Ciapponi A. 2020. Short-term exposure to particulate matter (PM10 and PM2.5), nitrogen dioxide (NO2), and ozone (O3) and all-cause and cause-specific mortality: Systematic review and meta-analysis. *Environment International*. 142, 105676. <https://doi.org/10.1016/j.envint.2020.105876>

*Response: This meta-analysis concern short-term effects and have therefore not been cited in this work.

R2.4. The authors have done a lot of work. Is there a way forward? The authors could use their results, Figure 2a-c, Figure 3a-c, figure S1a-c, Figure S2a-c, etc., to present a much better statistical picture of their results. Each estimated risk ratio and confidence limits can be used to compute a p-value. If you order the p-values from smallest to largest, they can be plotted against the integers, 1, 2, 3, ... to give a p-value plot. The reader can see all the statistical test in the context of the other tests in one figure. Also, the p-values can be used to produce multiplicity adjusted p- values so that the reader can judge the statistical reliability of the individual p-values/claims.

Supplemental Information for meta-analysis evaluation

S. Stanley Young, CGStat, Raleigh NC 27607

Warren Kindzierski, School of Public Health, University of Alberta, Edmonton, Alberta T6G 1C9. See arXiv 1808.04408-1.

I think this addition to the paper would help the reader evaluate the work. If the authors choose to go in this direction, I am willing to help.

*Response: Such adjustments for the familywise error rate are not typically used in this area of research except when the analysis concerns further investigation of subgroups which then repeatedly test the same hypothesis. Increased hazard ratio estimates were observed for both natural and CVD mortality in relation to PM10, PM2.5 and BC. P-values were not presented but 95% confidence intervals were calculated to present the statistical uncertainty of the hazard ratios.

Specific comments

R2.5. Abstract, page 2, line 31

In the abstract the authors say, "These findings are relevant for future decisions concerning air quality policies." Two points: Until this research area deals adequately with multiple testing and multiple modeling, research claims are unreliable. Recently the US EPA, in the interest of research transparency, want public access to data used in key papers used for regulatory decisions.

*Response: The study is relevant for future decisions concerning air quality policies since it contributes with hazard ratios of associations in populations with relatively low exposure to particulate air pollution. We agree that public access to data is important for transparency but do not disregard the scientific contribution from studies where for instance ethical permissions do not allow to share data.

R2.6. Page 4, line 3/6

Enstrom 2017 gives a meta-analysis of cohort studies and finds no effect. Young and Kindzierski 2019 examine a meta-analysis that appeared in JAMA and find no mortality effects on heart attacks. Unless authors deal effectively with multiple testing and multiple modeling their claims are unreliable for serious regulatory decisions.

Young SS, Kindzierski KB. 2019. Evaluation of a meta-analysis of air quality and heart attacks, a case study, *Critical Reviews in Toxicology*, doi: 10.1080/10408444.2019.1576587

*Response: The studies on short-term effects is not used as a basis for this paper.

R2.7. Page 6, line 15/10

The authors examine PM10, PM2.5, and black carbon. They should note if other air components were available and were they examined?

*Response: Source fractions, such as vehicle exhaust and road dust, were available for PM10 and PM2.5 but were highly correlated, and also correlated with total PM2.5 and PM10, respectively.

Including these source fractions would mainly increase the number of exposure variables, with poor chances to identify significant differences in effects because of their correlations.

R2.8. Page 6, line 28/20

It is common to use statistical significance of 0.05 with no correction for multiple testing or multiple modeling in environmental epidemiology. Until there is better statistical control of false positives, authors and readers should keep in mind the statistical reliability of these studies. The authors should comment.

Young SS. (2017) Air quality environmental epidemiology studies are unreliable. *Regulatory Toxicology and Pharmacology* 88, 177-180.

*Response: Please see our response to R2.4.

R2.9. Page 6, line 38/28

The predictor variables, PM10, PM2.5, and black carbon, are highly correlated among themselves and likely over time. The authors should provide somewhere a correlation matrix.

*Response: The correlations have been added (page 8, lines 27-29).

R2.10. Page 6, line 41/30

The words ESCAPE project appear in 93 titles of journal papers. It appears that data from this project has been used to address many questions.

*Response: The ESCAPE project has in several papers estimated the long-term effects on human health of exposure to air pollution in Europe.

Page 7, line 9/5

This paper is dependent on the reliability of the dispersion model, among many other things. I'm not familiar with this model.

*Response: The models used are state-of-the-art and have been validated against measurement data (validations were presented on page 7 line 11).

R2.11. Page 8, line 12/8.

I suspect that Ethical Review Boards are not familiar with multiple testing/multiple modeling, so they do not raise the MTMM issue with researchers.

*Response: The Ethical review boards have high scientific competence in epidemiological methods and it is the authors' experience that multiple testing is being discussed.

R2.12. Page 8, line 25/17ff.

The authors note that they had access to many other health effects.

*Response: Associations with incident ischemic heart disease and stroke have previously been published (page 9, line 29; Ljungman et al., 2019).

Reference:

Ljungman PLS, Andersson N, Stockfelt L, et al. Long-Term Exposure to Particulate Air Pollution, Black Carbon, and Their Source Components in Relation to Ischemic Heart Disease and Stroke. *Environmental Health Perspectives* 2019;127(10):107012. doi: 10.1289/EHP4757

R2.13. Page 9, line 20/14.

Confounding variables present a problem for observational studies. Yes, researchers would like to statistically remove possible biases and yes, they can be used to decrease the experimental error, BUT they can lead to try this and try that until an effect is found. And, important confounding variables might not be included. For example, radon is an important confounder for lung cancer. See *N Engl J Med* 1994; 330:159-164 DOI: 10.1056/NEJM199401203300302

“The estimates of risk were in the same range as those projected from data in miners. The interaction between radon exposure and smoking with regard to lung cancer exceeded additivity and was closer to a multiplicative effect.”

*Response: The confounding variables included were a priori selected based the recent literature. Radon exposure at levels occurring in dwellings is not conclusively related to any other disease or cause of death than lung cancer. Consequently, radon exposure is irrelevant for almost all analyses in the manuscript since lung cancer constitutes only a small fraction of overall mortality. As for the lung cancer specific analyses, negative confounding by radon exposure may be expected, if anything (see reference cited by reviewer). This is because radon levels tend to be lower in urban areas, where air pollution levels are higher, as fewer dwellings are close to the ground, which is the dominant source of indoor radon.

R2.14. Page 10, line 31/22

These results suggest that low dose results are largely from Umea. Location and dose appear to be confounded.

*Response: That is correctly understood. Hazard ratios were however estimated separate for each cohort, and thereafter combined in a meta-analysis.

R2.15. Page 11, lines 28/20

The health effects of stroke, ischemic heart disease are mentioned. It would help the reader to know about claims made using one or more of the cohorts to better get an idea of the extent of multiple testing and multiple modeling.

*Response: We believe that these were presented on page 9 lines 29-31.

R2.16. Page 12, lines 11/7

It seems clear to me that none of the cited papers adjusted for multiple testing or multiple modeling.’

*Response: That is correct

R2.17. Page 12, lines 20/14

None of the predictor variables, PM10, PM2.5, and black carbon, are chemically defined, so PM2.5 in one location might be radically different from PM2.5 in another locations.

*Response: The sources of PM10, PM2.5 and BC for these three cities have previously been described by Segersson et al. (2017).

Reference:

Segersson D, Eneroth K, Gidhagen L, et al. Health Impact of PM10, PM2.5 and Black Carbon Exposure Due to Different Source Sectors in Stockholm, Gothenburg and Umea, Sweden. *International Journal of Environmental Research and Public Health* 2017;14(7) doi: Artn 742 10.3390/ijerph14070742

R2.18. Page 13, Lines 22/15ff. Conclusions.

With a p-value plot and multiplicity adjusted p-values these conclusions should be reexamined.
 *Response: The study implemented the same approach to multiple testing as within previous studies. The study showed increased risks of mortality in relation to particulate matter exposure. Since these risk increases were observed in areas with relatively low concentrations of particulate matter air pollution we believe that the findings are relevant for future decisions concerning air quality policies.

VERSION 2 – REVIEW

REVIEWER	Whanhee Lee Yale University
REVIEW RETURNED	16-Feb-2021

GENERAL COMMENTS	Thank you for your great works. I think this manuscript is ready to be published.
---

REVIEWER	Stanley S. Young CGSTAT
REVIEW RETURNED	26-Feb-2021

GENERAL COMMENTS	General comments These researchers hold to their views air pollution is a killer and dismiss any concerns of multiple testing and multiple modeling. They appear to agree that claims of death in time series studies are not valid to dismiss that literature as applicable to cohort studies. Now if current poor air quality is not an acute killer strong evidence is needed to assert it is a chronic killer. Only one of the three cited cohort studies “statistically significant”. Let’s discuss multiple testing and multiple modeling, MTMM. The authors agree that there are very many, hundreds?, questions at issue in the cohort studies. They dismiss any concerns with MTMM with the off-hand remark that no one else is correcting for MTMM. They need to look at RCT literature AND take it seriously. The authors have not addressed my concerns R2.1. They look at three air quality measures, PM10, PM2.5, and black carbon. They look at multiple health measures, Natural mortality (all-cause), respiratory disease, cardiovascular disease, and lung cancer. They note that they have access to a full ranged of causes of deaths. They examine time windows of 1-5 years, 6-10 years, and a lag of 6-10 years. For exposure they mathematical dispersion model. A lot of questions are examined, air quality 3; health measures 4+; time periods 3+; demographic covariates, 8+. So, there are at least $2 \times 4 \times 3 \times 8 = 192$ questions/models at issue. All testing is done without any adjustment for multiple testing or multiple modeling, MTMM. *Response: The authors would like to note that this is not the number of hypotheses under study nor
--

	the number of models fitted but rather the total number of parameters estimated. **R2: I find no record of an analysis protocol being filed. I do note that the authors acknowledge many papers having been written about these cohort studies. One paper per question is fine, but an overarching treatment of MTMM is nowhere to be found. R2.2. None of the literature cited to help evaluate/support their claims does anything to address multiple testing or multiple modeling. Two of the cohort studies used in this report are part of the larger studies, Primary Prevention Study and MONICA. A google scholar search of "MONICA project" returned just over one million hits.. *Response: These studies were not cited since they either estimate short-term effects or have methodological flaws that may to a large extent have biased the results. This critique has also been published (Brunekreef and Hoek, 2006; Gapstur and Brawley, 2017; Pope et al, 2017). **R2: The authors use appeal to authority when it is not necessary. Young et al. 2017 provided their data set in 2015. No one disputes its result. Enstrom 2017 provides a meta-analysis of cohort studies. RR and CIs are given. It is negative. MTMM is a ready explanation for positive results. A scientist making a claim must provide strong supporting evidence. So far as I know, none of the authors that dispute Enstrom 2005 or 2017 make any of their own data public. R2.3. A recent study funded by WHO gathered and examine many air quality/health effect papers. They used 27 papers to evaluate all-cause mortality. They give a meta-analysis risk ratio of 1.0065, which I consider small enough to call claims into question. Response: This meta-analysis concern short-term effects and have therefore not been cited in this work. **R2: Do the authors agree with the WHO funded study of essentially no short-term effect? Readers would like to know the opinion of the authors on short-term effects. It is relevant. R2.4: MTMM *Response: Such adjustments for the familywise error rate are not typically used in this area of research except when the analysis concerns further investigation of subgroups which then repeatedly test the same hypothesis. Increased hazard ratio estimates were observed for both natural and CVD mortality in relation to PM10, PM2.5 and BC. P-values were not presented but 95% confidence intervals were calculated to present the statistical uncertainty of the
--	---

	hazard ratios. **R2: Appeal to authority and custom makes no sense when the authority is itself wrong. The 95% CLs do not control for the multiplicity of the questions. There are lots of questions. 0.05 is not sufficient control no matter how commonly (self-servingly) it is used. No RCT would be allowed to get away with this level of statistical control. Authors: We agree that public access to data is important for transparency but do not disregard the scientific contribution from studies where for instance ethical permissions do not allow to share data. **R2: Scientific contribution?? Is it really science if no one can see the data. MTMM concerns alone call the contribution into question. I point to "Figure 2a-c, Figure 3a-c, figure S1a-c, Figure S2a-c, etc.," which the authors ignore. For every RR and pair of CIs , a p-value can be computed. R2.6. Page 4, line 3/6 original paper Enstrom 2017 gives a meta-anlysis of cohort studies and finds no effect. *Response: The studies on short-term effects is not used as a basis for this paper. **R2: This response is non-responsive. *Response: Please see our response to R2.4. "Such adjustments for the familywise error rate are not typically used in this area of research..." **R2: Just because hundreds of papers pay not attention to MTMM does not make it a valid procedure. No RCT could get away with ignoring multiple testing and multiple modeling. Ignoring MTMM is self-serving and does not help the reader. R2.7 Other air components. **R2: The authors do not say if other common air components were available. Common air components: CO, SO₂, NO₂, etc, the typical air components. Did they have access to these other components? Did they check them? and find no effect? R2.10. Page 6, line 41/30 The words ESCAPE project appear in 93 titles of journal papers. It appears that data from this project has been used to address many questions. *Response: The ESCAPE project has in several papers estimated the long-term effects on human health of exposure to air pollution in Europe. **R2: So, yes, the authors acknowledge that there are multiple papers. MTMM again. How many papers do you get to write on one data set while ignoring MTMM? Authors: The Ethical review boards have high scientific competence in epidemiological methods... R2: My experience in reading hundreds of epidemiology papers that VERY seldom are multiple testing and multiple models taken into
--	--

	account. Epidemiology studies are notoriously unreliable. To say that ethical review board "talk" about it, but actually do nothing speaks poorly of epidemiology and review boards. R2 comment: That other endpoints have been reported confirms that multiple testing is an issue. Environmental epidemiology appears to be taking the position that you can look at multiple questions and write multiple papers. The left hand does not know (or care) what the right hand is doing. Environmental Epidemiology is without statistical support. It comes across as a data dredge. Organized p-HACKing. And the journal, Environmental Health Perspectives is a leading part of the problem.
--	--

VERSION 2 – AUTHOR RESPONSE

General comments

These researchers hold to their views air pollution is a killer and dismiss any concerns of multiple testing and multiple modeling. They appear to agree that claims of death in time series studies are not valid to dismiss that literature as applicable to cohort studies. Now if current poor air quality is not an acute killer strong evidence is needed to assert it is a chronic killer. Only one of the three cited cohort studies "statistically significant".

Let's discuss multiple testing and multiple modeling, MTMM. The authors agree that there are very many, hundreds?, questions at issue in the cohort studies. They dismiss any concerns with MTMM with the off-hand remark that no one else is correcting for MTMM. They need to look at RCT literature AND take it seriously.

The authors have not addressed my concerns

R2.1. They look at three air quality measures, PM10, PM2.5, and black carbon. They look at multiple health measures, Natural mortality (all-cause), respiratory disease, cardiovascular disease, and lung cancer. They note that they have access to a full ranged of causes of deaths. They examine time windows of 1-5 years, 6-10 years, and a lag of 6-10 years. For exposure they mathematical dispersion model. A lot of questions are examined, air quality 3; health measures 4+; time periods 3+; demographic covariates, 8+. So, there are at least $2 \times 4 \times 3 \times 8 = 192$ questions/models at issue. All testing is done without any adjustment for multiple testing or multiple modeling, MTMM.

*Response: The authors would like to note that this is not the number of hypotheses under study nor the number of models fitted but rather the total number of parameters estimated.

**R2: I find no record of an analysis protocol being filed. I do note that the authors acknowledge many papers having been written about these cohort studies. One paper per question is fine, but an overarching treatment of MTMM is nowhere to be found.

***Response: The a priori specification of model covariates and associations that the study aimed to assess have now been added (row 45 page 10 – row 6 page 11).

R2.2. None of the literature cited to help evaluate/support their claims does anything to address multiple testing or multiple modeling. Two of the cohort studies used in this report are part of the larger studies, Primary Prevention Study and MONICA. A google scholar search of "MONICA project" returned just over one million hits..

*Response: These studies were not cited since they either estimate short-term effects or have methodological flaws that may to a large extent have biased the results. This critique has also been published (Brunekreef and Hoek, 2006; Gapstur and Brawley, 2017; Pope et al, 2017).

**R2: The authors use appeal to authority when it is not necessary. Young et al. 2017 provided their data set in 2015. No one disputes its result. Enstrom 2017 provides a meta-analysis of cohort studies. RR and CIs are given. It is negative. MTMM is a ready explanation for positive results. A scientist making a claim must provide strong supporting evidence. So far as I know, none of the authors that dispute Enstrom 2005 or 2017 make any of their own data public.

*****Response:** The scientific evidence of air pollution effects is judged based on all published studies. Data can not always be shared due to limitations in the ethical approval. Enstrom (2017) was not cited since methodological deficiencies (including absence of advanced modeling approaches for exposure assessment and relatively limited number of individual-level covariates and does not control for any ecologic covariates) may to a large extent have biased the results. This critique has also been published (Pope et al, 2017).

Reference:

Pope CA, 3rd, Krewski D, Gapstur SM, et al. Fine Particulate Air Pollution and Mortality: Response to Enstrom's Reanalysis of the American Cancer Society Cancer Prevention Study II Cohort. *Dose Response* 2017;15(4):1559325817746303. doi: 10.1177/1559325817746303 [published Online First: 2017/12/26]

R2.3. A recent study funded by WHO gathered and examine many air quality/health effect papers. They used 27 papers to evaluate all-cause mortality. They give a meta-analysis risk ratio of 1.0065, which I consider small enough to call claims into question.

Response: This meta-analysis concern short-term effects and have therefore not been cited in this work.

**R2: Do the authors agree with the WHO funded study of essentially no short-term effect? Readers would like to know the opinion of the authors on short-term effects. It is relevant.

*****Response:** The amount of evidence for short term effects is also large. The authors find the conclusion of the review and meta-analysis by Orellano et al. (2020) to be well supported:

"This study found evidence of a positive association between short-term exposure to PM10, PM2.5, NO2, and O3 and all-cause mortality, and between PM10 and PM2.5 and cardiovascular, respiratory and cerebrovascular mortality. These results were robust through several sensitivity analyses. In general, the level of evidence was high, meaning that we can be confident in the associations found in this study."

Since the study is on short-term effects is not cited within the current work.

R2.4: MTMM

*Response: Such adjustments for the familywise error rate are not typically used in this area of research except when the analysis concerns further investigation of subgroups which then repeatedly test the same hypothesis. Increased hazard ratio estimates were observed for both natural and CVD mortality in relation to PM10, PM2.5 and BC. P-values were not presented but 95% confidence intervals were calculated to present the statistical uncertainty of the hazard ratios.

**R2: Appeal to authority and custom makes no sense when the authority is itself wrong. The 95% CIs do not control for the multiplicity of the questions. There are lots of questions. 0.05 is not

sufficient control no matter how commonly (self-servingly) it is used. No RCT would be allowed to get away with this level of statistical control.

*****Response:** This study does not repeatedly test the same hypothesis. It does however estimate associations with three measures of exposure during two different time windows. This has now been clarified in the discussion (row 45 page 10 – row 6 page 11).

Authors: We agree that public access to data is important for transparency but do not disregard the scientific contribution from studies where for instance ethical permissions do not allow to share data.

****R2:** Scientific contribution?? Is it really science if no one can see the data. MTMM concerns alone call the contribution into question. I point to “Figure 2a-c, Figure 3a-c, figure S1a-c, Figure S2a-c, etc.,” which the authors ignore. For every RR and pair of CIs, a p-value can be computed.

*****Response:** We believe that the results of these studies add to the scientific knowledge. A clarification concerning the familywise error rate have now been added (row 45 page 10 – row 6 page 11).

R2.6. Page 4, line 3/6 original paper

Enstrom 2017 gives a meta-analysis of cohort studies and finds no effect.

*Response: The studies on short-term effects is not used as a basis for this paper.

****R2:** This response is non-responsive.

*****Response:** See our response above: “Enstrom (2017) was not cited since methodological deficiencies (including absence of advanced modeling approaches for exposure assessment and relatively limited number of individual-level covariates and does not control for any ecologic covariates) may to a large extent have biased the results. This critique has also been published (Pope et al, 2017).”

Reference:

Pope CA, 3rd, Krewski D, Gapstur SM, et al. Fine Particulate Air Pollution and Mortality: Response to Enstrom's Reanalysis of the American Cancer Society Cancer Prevention Study II Cohort. *Dose Response* 2017;15(4):1559325817746303. doi: 10.1177/1559325817746303 [published Online First: 2017/12/26]

*Response: Please see our response to R2.4. “Such adjustments for the familywise error rate are not typically used in this area of research...”

****R2:** Just because hundreds of papers pay not attention to MTMM does not make it a valid procedure. No RCT could get away with ignoring multiple testing and multiple modeling. Ignoring MTMM is self-serving and does not help the reader.

*****Response:** See our response above: “This study does not repeatedly test the same hypothesis. It does however estimate associations with three measures of exposure during two different time windows. This has now been clarified in the discussion (row 45 page 10 – row 6 page 11).”

R2.7 Other air components.

****R2:** The authors do not say if other common air components were available. Common air components: CO, SO₂, NO₂, etc, the typical air components. Did they have access to these other components? Did they check them? and find no effect?

*****Response:** These gases were not part of our hypothesis and thus not modeled.

R2.10. Page 6, line 41/30

The words ESCAPE project appear in 93 titles of journal papers. It appears that data from this project has been used to address many questions.

*Response: The ESCAPE project has in several papers estimated the long-term effects on human health of exposure to air pollution in Europe.

**R2: So, yes, the authors acknowledge that there are multiple papers. MTMM again. How many papers do you get to write on one data set while ignoring MTMM?

*****Response:** The period of follow-up differs between these studies, also the exposure modeling and assessment, which were in the ESCAPE-studies only performed for the year of inclusion.

Authors: The Ethical review boards have high scientific competence in epidemiological methods...

R2: My experience in reading hundreds of epidemiology papers that VERY seldom are multiple testing and multiple models taken into account. Epidemiology studies are notoriously unreliable. To say that ethical review board "talk" about it, but actually do nothing speaks poorly of epidemiology and review boards.

*****Response:** The methodological limitations of observational studies have now been added to the discussion (row 45 page 10 – row 6 page 11).

R2 comment: That other endpoints have been reported confirms that multiple testing is an issue. Environmental epidemiology appears to be taking the position that you can look at multiple questions and write multiple papers. The left hand does not know (or care) what the right hand is doing.

*****Response:** Associations with IHD and stroke incidence has previously been published (as mentioned in the discussion). We do not however repeatedly test the same hypothesis.

Environmental Epidemiology is without statistical support. It comes across as a data dredge. Organized p-HACKing. And the journal, Environmental Health Perspectives is a leading part of the problem.